# Gastroesophageal Neuroendocrine Tumors: Outcomes and Management

**DOI:** 10.3390/jcm14072148

**Published:** 2025-03-21

**Authors:** Christine Son, Joshua Kalapala, Jeff Leya, Michelle Marion Popadiuk, Mohammed K. Atieh, Daniel Havlichek, Lawrence Feldman, Paul Roach, Promila Banerjee

**Affiliations:** 1Edward Hines VA Hospital, Hines, IL 60141, USA; jeff.leya@va.gov (J.L.); michelle.popadiuk@va.gov (M.M.P.); mohammed.atieh@va.gov (M.K.A.); lawrence.feldman@va.gov (L.F.); paul.roach@va.gov (P.R.); promila.banerjee@va.gov (P.B.); 2Loyola Internal Medicine, University Medical Center, Maywood, IL 60153, USA; 3Stritch School of Medicine, Maywood, IL 60153, USA; joshua.kalapala@va.gov; 4UI Health, University of Illinois in Chicago Hospital Health Sciences System, UIC Medical Center, Chicago, IL 60612, USA

**Keywords:** neuroendocrine tumors, gastroesophageal neuroendocrine tumors, neuroendocrine tumor outcomes

## Abstract

**Background/Objectives**: Neuroendocrine tumors (NETs) can arise in any organ and are most commonly found in the lungs and gastroenteropancreatic (GEP) system. GEP-NETs represent a small percentage of gastrointestinal cancers, and therefore, the standard treatment is not well-defined, especially for advanced disease. Our objective is to review GI NETs among veterans and analyze their therapeutic outcomes. **Methods**: A total of 61 GI NET cases were identified from our institution from 2019–2024. In total, twenty-seven review papers, ten population-based/multicenter/outcome studies, six case reports, and one case series were reviewed for the literature review. **Results**: The incidence of GI NETs at our institution was higher than the known epidemiology of GI NETs. Small intestine NETs were one of the most common sites of GEP-NETs at our institution, with only one of nineteen cases being grade 3 poorly differentiated neuroendocrine carcinoma. All cases of colonic and rectal NETs had good clinical outcomes consistent with findings from the literature. Most of the gastric NETs were type 1 and had benign courses of disease, except for one case with an intermediate grade and metastatic liver lesions. One case of esophageal neuroendocrine carcinoma (E-NEC) showed a complete response to chemotherapy despite a significant tumor burden on presentation and high-grade pathology, while another case of ENEC had recurrent disease despite systemic therapy. **Conclusions**: While the role of surgery or endoscopic resection is limited to localized tumors, combined treatment with chemoradiation can significantly improve patient outcomes, especially in high-grade, poorly differentiated tumors. Further studies are needed to establish systemic (i.e., chemotherapy and radiation) treatment strategies for poorly differentiated GI NETs.

## 1. Introduction

Gastroenteropancreatic neuroendocrine carcinomas (GEP-NECs) comprise less than 1% of gastrointestinal cancers [1]. Only 1.6% of all newly diagnosed NECs are found in the esophagus [2], and 5–15% are found in the stomach [3]. Neuroendocrine tumors (NETs) or carcinoids refer to grade 1, well-differentiated neuroendocrine neoplasms, while neuroendocrine carcinomas (NECs) refer to grade 3, poorly differentiated neuroendocrine neoplasms [1]. A higher incidence of GI NETs is seen in males than females, and it is more commonly seen in Blacks than Whites and other races [4]. The majority of esophageal NECs (E-NECs) show aggressive behavior, with less than 1% being well-differentiated [5]. E-NECs are generally of small-cell and large-cell variants, but the small-cell type is more frequent, accounting for 90% of total cases [6]. Gastric NETs (G-NETs), also known as gastric carcinoids, on the other hand, are categorized into three types, with each type having a distinct pathogenesis and prognosis [7,8]. Type 1 G-NETs are the most frequent (80–90%), followed by type 3 (10–15%) and type 2 (5–7%) [7].

NECs of the GI tract have a median survival of 4 to 15 months depending on the primary site and disease stage [2,5]. General guidelines suggest surgical intervention with adjuvant chemoradiotherapy as the treatment of choice for locally advanced or metastatic E-NECs. For G-NETs, treatment options include surveillance with or without excision, endoscopic or surgical resection, and systemic therapies [7]. The majority of the E-NECs show aggressive behavior, with poor overall prognosis, while the prognosis of G-NETs varies significantly [7]. In this study, we aim to review the treatment patterns and outcomes of GI NETs of luminal organs at our institution compared to the current literature. We aim to identify the features that are unique to our patient populations and provide the role of different treatment therapies for different tumor subtypes.

## 2. Materials and Methods

References for the literature review were identified through PubMed searches. In PubMed, the following search terms were used: “Neuroendocrine Tumors”, “Neuroendocrine Carcinomas”, “Colon Neuroendocrine Tumor”, “Rectal Neuroendocrine Tumor”, “Small Bowel Neuroendocrine Tumor”, “Small Bowel Neuroendocrine Tumor Treatment”, “Small Bowel Neuroendocrine Tumor Prognosis”, “Esophageal Neuroendocrine Carcinoma”, “Esophageal Neuroendocrine Carcinoma Prognosis”, “Esophageal Neuroendocrine Carcinoma Treatment”, “Gastric Neuroendocrine Tumors”, “Gastric Neuroendocrine Tumor Outcomes”, and “Gastric Neuroendocrine Tumor Therapeutic Outcomes”. In total, 27 review papers, 10 population-based/multicenter/outcome studies, 6 case reports, and 1 case series were reviewed for the literature review.

The Edwards Jr Hines VA Hospital Cancer Registry Database, which identifies all GI system tumors from 2019 to 2024, revealed a total of 723 GI system cancers. Of these 723 cancers, 61 (8.4%) were neuroendocrine tumors. Secondary or metastatic neuroendocrine tumors were included. For the purpose of this review article, we focused on analyzing the GI NETs of luminal organs.

## 3. Results

It comprised 20 pancreatic (32.8%), 19 small intestinal (31%), 9 rectal (14.8%), 9 gastric (14.8%), 2 colon (3.3%), and 2 esophageal NET (3.3%) cases. The terms neuroendocrine tumor and carcinoids were used interchangeably. (Figure 1). All diagnoses were made with pathology that included immunohistochemical stains performed on cell blocks for synaptophysin, chromogranin, CD-56, Ki-67 proliferation index, and mitosis count.

Excluding GI NETs, 662 total GI tumors were distributed in the following primary sites: 149 colon (22.5%), 88 esophagus (13.3%), 23 cholangiocarcinoma (3.5%), 120 liver (18.1%), 107 pancreas (16.2%), 15 biliary (2.3%), 69 rectum (10.4%), 69 stomach (10.4%), 6 small intestine (0.9%), 1 unspecified site of primary tumor (0.1%), and 15 anus (2.3%) (Figure 2).

The trend of individual GI NETs had no particular pattern, with varying numbers of incidences each year (Figure 3).

Overall, the highest number of tumors were found in the colon, followed by the liver and pancreas with a steady pattern of incidence. The overall trend was steady in most organ sites (Figure 4).

GEP-NETs are classified as low-, intermediate-, or high-grade tumors based on the mitotic index and Ki-67 proliferation index, which refers to the rate at which the tumor cells divide. Grade 1 refers to a Ki-67 index of 2% or lower. Grade 2 refers to a Ki-67 index between 3 and 20%. Grade 3 refers to a Ki-67 index higher than 30% [7].

All NETs are staged according to the tumor (T), node (N), and metastasis (M) staging system, and the prognosis is dependent on the degree of regional lymph node metastasis, tumor size, and pathological stage [9].

### 3.1. Gastric NETs

Eight out of nine gastric NETs were classified as type 1, with either elevated gastrin levels or a history of atrophic gastritis supported by EGD findings. None of the G-NETs were found to be a gastrinoma or Zollinger–Ellison syndrome. (Table 1.) While not delineated specifically, one case of G-NET required radical resection, including partial or total gastrectomy, given the multifocal tumor burden, higher grade, or evidence of metastases at the time or presentation, which is suspicious for a type 3 G-NET.

All nine gastric NETs at our institution were well-differentiated with grades no greater than 2. Eight out of nine cases were classified as type 1 with elevated gastrin levels and EGD findings of atrophic gastritis—“gastric mucosa with moderate chronic active gastritis and intestinal metaplasia”. All patients with type 1 gastric NET survived except one patient who passed due to another comorbidity. Four patients underwent endoscopic resection and were followed annually with surveillance EGD. One patient underwent subtotal gastrectomy with Roux-En-Y reconstruction given the tumor size (2.3 × 1.8 × 0.9 cm), intermediate grade (Ki-67 proliferation index of 12%), invasion of the submucosa, and multifocal stage 2 disease despite the presence of atrophic gastritis. Given no evidence of disease on the Dotatate PET scan, the patient did not undergo further adjuvant therapy and continued with surveillance imaging every 3 months for 1 year and then annually (Table 2).

One patient with a type 1 G-NET was initially found to have a stage 2b gastric adenocarcinoma after a syncopal episode following melena. Subsequent CT imaging found a 3.8 × 2.5 cm lobulated mass on the distal body of the stomach, abutting the left liver lobe, without evidence of lymphadenopathy or metastasis. The patient underwent perioperative chemotherapy and eventually robot-assisted distal gastrectomy with Roux-en-Y reconstruction, and the distal gastric pathology revealed a well-differentiated NET that was 0.2 mm in size and a low Ki-67 index of 1–2%.

Another case of a type 1 G-NET was incidentally discovered with a gastric polyp upon a diagnostic EGD for iron deficiency anemia. The pathology showed a well-differentiated grade 2 NET, and the patient continued with annual surveillance EGD.

While not delineated clearly, one case of a G-NET was of intermediate grade [Ki-67 index of 4–5%] and presented with multiple liver lesions with abdominal mesentery and lymphadenopathy with several sub-centimeter metabolic lung nodules. The right middle lobe biopsy showed a de-differentiated metastatic NET. The patient’s chromogranin-A level was noted to be elevated in the 5000s–8000s, although it was deemed unreliable as the patient was on PPI. Serum serotonin levels were also noted to be high in the 1000s and 24 h urine 5-HIAA was elevated at 43. The patient was initially treated with octreotide as well as transarterial chemoembolization (TACE) for the liver lesions. Upon surveillance imaging, disease progression was noted, and the patient was started on everolimus and eventually lutathera, a lutetium oxodotreotide.

### 3.2. Esophageal NETs

Two cases of esophageal NETs were found at our institution. Both patients presented with worsening solid food dysphagia and were found to have a large esophageal mass that was concerning for primary esophageal malignancy on CT imaging. Tissue pathology for both showed high-grade, neuroendocrine carcinomas with hypermetabolic lymph nodes near the primary lesion. The two cases were treated similarly given the advanced disease at presentation but produced different clinical outcomes to date (Table 3).

In the first case of ENEC, the patient had a good initial response after concurrent chemoradiation with cisplatin/etoposide. However, the patient was noted to have a recurrence upon a subsequent EGD 11 months after the completion of chemoradiotherapy. No obvious new metastases were seen. The patient was then treated with ipilimumab and nivolumab but continued to have progressive disease. The patient then completed two cycles of carboplatin/etoposide but was unable to continue treatment due to multiple complications, including a prostatic abscess, ischemic stroke, and failure to thrive in a setting of worsening dysphasia, likely with disease progression. The patient was transitioned to hospice care and passed within 2 years of the initial diagnosis.

The second case of ENEC in 2023 presented with a large friable mass with food components on the initial outpatient EGD [10]. The procedure was aborted as the mass could not be traversed with oozing blood and a blood clot in the stomach. The patient was intubated for airway protection and transferred to our tertiary hospital for a higher level of care. Upon transfer, CT imaging showed a large esophageal mass extending into the gastric cardia and measuring 6.7 cm × 6.0 cm × 8.8 cm without evidence of perforation. On a subsequent EGD, a fleshy hemorrhagic mass with food particles was seen in the distal 15–20 cm of the esophagus. There was stenosis of the gastroesophageal junction and retained blood clots and food debris prevented adequate evaluation of the distal esophagus. An overtube placement with repeated passes with a cold snare revealed a friable, soft tissue. The degree of tumor mass compression on the posterior left atrium was significant, causing recurring episodes of supraventricular tachycardia. An esophageal stent was not placed due to concerns that further compression of the left atrium could worsen cardiac function and the high likelihood of stent migration. The patient further developed lower extremity DVTs and pulmonary embolism requiring anticoagulation. Given numerous complications, the surgery was deferred. Ultimately, the patient completed definitive chemoradiation with four cycles of cisplatin/etoposide, with three post-treatment PETs showing no evidence of disease. The patient will continue to be monitored with surveillance PET scans with EGDs and proceed with salvage esophagectomy if the disease recurs.

### 3.3. Small Intestinal NETs

Amongst the 19 cases of small bowel NETs, 4 cases were found in the terminal ileum, 4 in the duodenal bulb, 3 in the distal ileum, 1 in the jejunum and ileum, and 1 in the ileocecum. The remaining five cases of NETs presented with multifocal lesions (Table 4).

Two incidental tumors found on the duodenal bulb on a diagnostic EGD for indications of iron deficiency anemia and bloating were found to have low-grade, well-differentiated NETs with no lymph node invasions or distant metastasis. A repeat EGD one year later showed no recurrence of disease and a surveillance EGD was not recommended unless symptoms recur.

The remaining cases all presented with distant metastases, with the liver being the most common site of spread. Two cases of small bowel NETs with liver metastases were either grade 1 or 2, well-differentiated NETs. Only one case was diagnosed as a poorly differentiated, grade 3 NEC with metastatic disease to the liver.

Surgical resection was the first line of treatment and somatostatin analogs, such as octreotide and lanreotide, were used to suppress tumor growth. Everolimus, a targeted drug that blocks the mTOR kinase protein, was used when the patient had persistent disease progression despite treatment with somatostatin analogs. Patients were followed with serial imaging with CT or MRI with chromogranin A measures every 3 months post-operatively (Table 5).

A patient with a grade 3 NEC was treated with adjuvant cisplatin/etoposide initially, followed by monthly octreotide and surveillance. However, the patient was noted to have a progression of hepatic disease and additionally was found to have a stage 2 colon adenocarcinoma with lymph node involvement. No further adjuvant therapy was offered due to a high-grade NET or NEC, and the patient passed within 2 years of the initial diagnosis.

Worsening ascites were the main presenting symptom for one patient who was found to have a grade 2, well-differentiated NET of the terminal ileum with metastatic disease to the omentum. CT imaging of the abdomen and pelvis also demonstrated three small subpleural nodules of up to 1.6 cm in size, thickening of both small and large bowel walls, which was concerning for tumor infiltration, and mesenteric and omental thickening, which was concerning for carcinomatosis. A biopsy further demonstrated a Ki-67 index of up to 10% and the tumor was found to have infiltrated the lamina propria, muscularis mucosae, and apparent submucosa. The patient was started on octreotide for target therapy to slow tumor growth and a pleurx peritoneal drainage catheter was placed for recurrent ascites. The patient passed within 5 months of the initial presentation.

A metastatic ileocecal NET was also found to have peritoneal involvement, and the patient was able to undergo a right hemicolectomy with ileocolic anastomosis. However, the patient continued to have a residual functional NET with persistently elevated chromogranin A and 5-HIAA levels and was started on octreotide.

### 3.4. Colonic NETs

Two cases of colon carcinoids were found incidentally following an appendectomy. Both were stage 1, well-differentiated NETs. Both were small, 0.3 cm and 0.2 cm in maximum dimension, and surgical margins were negative for malignancy. No other treatment was pursued for both cases. Routine surveillance imaging was deemed not necessary (Table 6).

### 3.5. Rectal NETs

Five cases of rectal NETs were found during routine screening colonoscopies. Three cases of rectal NETs were found during colonoscopies performed for other indications, including chronic dyspepsia, anemia, and polyp surveillance.

One patient had a known diagnosis of sigmoid adenocarcinoma and was found to have a rectal well-differentiated NET that was 0.6 cm in its max dimension. Once tissue pathology returned with a neuroendocrine tumor, a rectal EUS and flexible sigmoidoscopy were performed to look for evidence of recurrence.

Post-treatment surveillance was not recommended for tumors <1 cm in size given the exceptionally low risks for recurrence. Most patients were continued with routine colonoscopy surveillance if the original NET was <1 cm in size and had good clinical outcomes (no recurrence of disease) (Table 7).

All colonic NET and rectal NET patients had good clinical outcomes except one colonic NET patient who passed from old age. Three out of nineteen small intestine NET patients passed from recurrent, metastatic disease but had varying lengths of survival (4 to 24 months). One small intestinal NET patient passed from another metastatic disease and one patient passed from other multiple comorbidities. One out of the nine gastric NET patients passed from another comorbidity. One of the two ENET patients passed from recurrent disease. Recurrence of disease was defined with the disease burden seen on surveillance imaging. Rarely, patients presented with new symptoms, and elevations in biomarkers were routinely followed only for certain patient cases (Table 8).

## 4. Discussion

### 4.1. Gastric NETs

Although rare, the incidence of gastric neuroendocrine tumors has been increasing over the years [11]. Gastric NETs (G-NETs) are divided into three types according to clinical features [7]. Type 1, or autoimmune atrophic gastritis, typically has a more favorable outcome with lower metastatic potential [7]. Type 2 refers to Zollinger–Ellison syndrome or gastrin-releasing tumor, and surgical management is often required [7]. Type 3 gastric NETs are often sporadic and usually consist of a unifocal lesion with normal gastric levels. However, they are associated with a high level of metastasis [7]. It is crucial to identify the type of G-NET because management, treatment, and prognoses differ significantly between subtypes.

Type 1 G-NETs are often treated with endoscopic resection in the absence of metastatic disease, and surveillance is tailored to the size and number of prominent lesions [7]. Somatostatin analogs can be used in the setting of recurrence following endoscopic resection or multiple lesions not amendable to endoscopic resection to reduce serum gastrin [7]. Adjuvant chemotherapy is reserved only for poorly differentiated G-NETs [10]. The use of immunotherapy or immune checkpoint inhibitors (ICIs) to treat GEP-NETs is still in the clinical exploration phase and is not yet recommended as the preferred regimen [12].

A recent study of 57 patients showed that 30% of patients monitored with surveillance required resection. In total, 50% of those patients required reinterventions, with a median follow-up of 22 months [13]. Another study followed 84 patients after endoscopic or surgical intervention and found that 52% developed a local recurrence requiring reintervention during a mean follow-up period of 45 months [14]. Additionally, according to one outcome study that looked at 125 patients who underwent resections of type 1 gastric NETs, the disease-free survival rate did not differ significantly between those who underwent endoscopic mucosal resection and those who underwent submucosal dissection, with local recurrence rates of 6.5% and 2.4%, respectively [8].

Type 3 G-NETs, representing 10–15% of G-NETs, are often at an advanced stage at the time of diagnosis. The European Neuroendocrine Tumor Society and NANETS recommend radical resection, including partial or total gastrectomy with regional lymphadenectomy, for non-metastatic lesions > 2 cm [7]. A small study that looked at patient outcomes of low-grade type 3 G-NETs without lymphovascular invasion found that only patients with tumor sizes > 1.5 cm developed recurrence [15].

Only one case of a likely type 3 G-NET was found at our institution, which showed a significant disease burden with multiple hypermetabolic lesions throughout the mesentery and retroperitoneum in the PET CT. Persistent diarrhea was the prominent symptom, and treatment was tailored to his symptoms with targeted therapies of lantreotide and lutathera. The patient was noted to have a progression of disease 26 months following the initial diagnosis and was started on everolimus. The patient was continued on lutathera treatment and is now 7 years out from the initial diagnosis.

### 4.2. Esophageal NETs

Comprising approximately 1.6% of esophageal cancers, cases of E-NECs are nonetheless on the rise [2]. A recent study found that 77% of E-NECs were found in the lower third of the esophagus, with a mean size of 2.3 cm at the time of diagnosis [16]. While most patients are asymptomatic at the time of presentation, approximately 25% present with dysphagia [17]. While high-grade tumors can grow quickly, near-complete esophageal obstruction with food bolus and hemorrhage remains a rare presentation, as seen in our patient in the latter case.

Less than 1% of E-NETs are well-differentiated, and 31–90% of cases present with common lymph node and distant metastases [18]. Distant metastases are commonly in the liver, lung, and bone [6]. Brain metastases are rare, with only a few cases from primary E-NECs reported in the literature [19].

Given the rarity of E-NECs, disease characteristics and standard treatment are not well-defined. General recommendations suggest surgical intervention with adjuvant chemoradiotherapy as the treatment of choice [5]. Due to similar histological manifestations and highly aggressive characteristics, small-cell carcinoma of the esophagus (SCCE) is treated similarly to small-cell lung cancer (SCLC).

Patients with tumors smaller than 2.0 cm in size appeared to have significantly better survival outcomes than those with tumors greater than 2.0 cm in size according to one meta-analysis [16]. However, there was no statistical significance regarding regional lymph node metastasis and lymphovascular invasion predicting better survival [16]. Inconsistent results were seen on whether synaptophysin or chromogranin expression is associated with a better prognosis [20]. Additionally, there was no difference in prognosis between large-cell NETs and small-cell NETs [21].

The survival outcomes of patients with locally advanced E-NECs treated with definitive chemoradiation therapy have not been well-studied, especially given the disease’s rarity. The most frequent combination of chemotherapy used was cisplatin and etoposide [16]. When comparing the clinical outcomes of locally advanced E-NECs treated with either platinum plus etoposide with radiotherapy (CE-RT) or cisplatin plus 5-fluorouracil with radiotherapy (CF-RT), the overall response rates and complete remission rates in all patients were 86.4% and 77.3%, respectively [22]. Patients treated with CE-RT experienced more hematological adverse events, including neutropenia and febrile neutropenia, but more long-term progression-free survival was observed than with CF-RT [22]. Overall survival was poor with rates of 37% at 1 year, 14% at 3 years, and 11% at 5 years [23].

A 2013 meta-analysis study of SCCE patients showed that chemotherapy was associated with approximately 30% 5-year survival compared to surgery alone, radiotherapy alone, or nonstandard and no therapy, at 15%, 24%, and 12%, respectively. Adding either surgery or radiotherapy to chemotherapy may provide additional benefits, but adding both did not provide further benefits [23].

One recent case report revealed that complete remission was achieved in advanced, metastatic E-NECs treated with tislelizumab combined with anlotinib as a second-line therapy [24]. This showed that immunotherapy, combined with targeted therapy, could represent a new treatment option for this disease. Treatment with peptide receptor radionuclide therapy (PRRT), such as lutathera for advanced, somatostatin receptor-expressing NETs has shown a prolonged time to progression, reduction of symptoms, and better quality of life [25]. Recent studies show that the surgical resection of locally advanced or metastatic NETs before PRRT can provide significant survival benefits with longer progression-free survival (18 months in the prior surgical resection group vs. 14 months in the non-prior resection group) and longer overall survival (134 months in the prior surgical resection group vs. 67 months) [26]. Such a favorable impact of PRRT following surgical resection is presumed to be due to lower tumor volumes in patients after surgery [27].

While distinguishing between small-cell and non-small-cell NET may not impact survival, it infers responsiveness to chemotherapy [9]. This distinction is well-known for lung cancers, but there is not enough data on its significance in NETs. In our surviving ENEC patient, histological findings of both large and small cells made morphologic diagnosis challenging. Because Ki-67 was positive in 60–70% of tumor cells, supporting a high proliferation fraction, it was categorized as a high-grade NET following the standard classification criteria [28].

### 4.3. Small Intestinal NETs

A small bowel NET is the most common primary site among GEP-NETs, following rectal NETs [29,30]. Up to 30% of patients present with metastatic disease, and those with liver metastases can present with carcinoid syndrome, with symptoms such as diarrhea, wheezing, and flushing [30]. This is due to the tumor secretion of serotonin, prostaglandins, and histamine as the byproducts of tumor secretion are usually metabolized by the liver [30]. The diagnosis of carcinoid syndrome can be aided by the collection of 5-HIAA, and imaging such as CT, MRI, and radiolabeled somatostatin analogs with gallium or indium penetetreotide can be used to determine the location and size of the disease [30].

The role of endoscopy in small bowel NETs continues to be studied [31]. For patients with localized small bowel NETs, which are typically stage 0, I, or II, the gold standard is surgical removal. This is done via segmental small bowel resection or an ileocecectomy [31]. The amount of lymph node involvement is the primary prognostic factor and the determinant of recurrence post-op [29]. However, the minimum number of lymph nodes that must be retrieved during surgery is not well-established in the literature [29]. Two retrospective studies that looked at patient outcomes following surgical resection showed that patients who underwent excision of mesenteric metastases had significantly longer survival than those with remaining lymph node metastases [32], and liver-directed surgery for those with liver metastases with hepatic resection, ablation, or both combined showed prolonged survival, with a 5-year survival rate of 74% and a 10-year survival rate of 51% [33].

While patients are rarely completely cured with surgery, when metastasis is present at the time of diagnosis, surgical management can increase survival time and improve their quality of life. One study that reviewed 706 transplanted metastatic small bowel NETs found that the 5-year survival rate of those who underwent a liver transplant was 70% compared to 34% for those who did not have the liver transplant [34]. However, the majority of patients still develop recurrent disease, with up to 95% of patients at 5 years and 99% at 10 years reported in one multicenter study [33]. Therefore, continued surveillance is important after surgical management. The current guidelines recommend 6-month surveillance visits for the first year, with an annual follow-up for 10 years [31]. Somatostatin analogs such as octreotide or lantreotide are first-line treatments for both functional and nonfunctional metastatic small bowel NETs [31]. Treatment with everolimus combined with somatostatin was found to improve progression-free survival in the RADIANT-2 trial, but there was no statistically significant difference between the two [31]. Currently, cytotoxic chemotherapy plays a smaller role in the treatment of well-differentiated NETs [31].

All small bowl NET cases with distant metastases had liver involvement at our institution, which is consistent with findings from the literature. Surgical resection was the first line of treatment and somatostatin analogs were used to suppress tumor growth. Everolimus was used when the patient had persistent disease progression despite treatment with somatostatin analogs, although it was difficult to tell if this added much to their prolonged survival as only one patient was treated with a combination of everolimus and a somatostatin analog. All patients with high-grade, metastatic disease passed within 5 to 24 months from the time of initial diagnosis. Additionally, these patients could not undergo surgical intervention due to significant disease burden on presentation, which contrasts with the high survival rates of those who underwent surgical resection reported in the literature. The only surviving patient with metastatic disease at the time of presentation had a grade 1, well-differentiated disease.

### 4.4. Colonic NETs

NETs of the appendix, colon, and rectum are mostly considered non-aggressive and benign, with only a few cases showing aggressive behavior [35]. Colonic NETs (C-NETs) make up 5–7% of all well-differentiated GEP-NETs with similar clinical presentations as adenocarcinoma of the colons [35]. Less than 1% of patients present with carcinoid features [i.e., flushing and diarrhea], usually in the setting of metastatic disease to the liver [36]. Appendiceal NETs constitute 16% of GI NETs [36], and most are incidentally found at the time of an appendectomy [36].

Most patients with small [<2 cm], well-differentiated appendiceal NETs without positive margins are considered low risk and no surveillance is needed [36]. If lesions are >2 cm, resection is incomplete, or there are positive margins, workup with a contrast-enhanced, triple-phase computed tomography [CT] scan or magnetic resonance imaging [MRI] is recommended [36].

According to the National Cancer Database, there was no survival benefit of pursuing a right hemicolectomy for patients with lesions 1.0 to 1.9 cm in size, even though the risk of nodal metastases is higher in this group compared to those with lesions < 1 cm [36]. Therefore, a right hemicolectomy is only recommended for lesions > 2 cm in most guidelines, and a simple appendectomy is considered sufficient therapy for lesions < 1 cm [36]. For lesions between 1.0 and 2.0 cm, a multidisciplinary approach is usually taken, which includes an assessment of high-risk features, the lesion site (base vs. apex), patient age and comorbidities, and the likelihood of surgical complications [37]. Additionally, if a right hemicolectomy is planned, a full colonoscopy is recommended to rule out concurrent colorectal cancers [36]. Overall, C-NETs have worse prognoses compared to rectal NETs [35].

This guideline corresponds to the management of two colonic/appendiceal NETs found at our institution. As both were small at 0.3 cm and 0.2 cm, with negative margins, no further work-up or routine surveillance was pursued. Similar to the literature review’s findings, all cases of colonic NETs were benign, with no recurrence to date.

### 4.5. Rectal NETs

Rectal NETs (R-NETs) comprise about 1–2% of all rectal tumors and represent 12–27% of all GI NETs [38]. Most rectal NETs are found incidentally during a screening colonoscopy, and the pathology is sent following a polypectomy [36]. There has been almost a 10-fold increase in the incidence of rectal NETs in the past 35 years with increased screening and the improved detection of colorectal cancers with colonoscopies [21].

Up to 90% of rectal NETs are well-differentiated, less than 10 mm in size, and do not invade the muscularis propria [36]. Localized lesions with diameters < 10 mm are treated with endoscopic mucosal resection (EMR) or endoscopic submucosal dissection (ESD) [37]. One meta-analysis in 2021 showed that complete resection rates were significantly higher in the ESD group compared with the EMR group [39]. For locally advanced R-NETs that invade submucosal layers and are >10 mm in size but without evidence of distant metastases, a rectal MRI and/or rectal endoscopic ultrasound (r-EUS) is recommended for staging [39]. Once distant metastases are excluded, surgical resection, such as low anterior resection (LAR) with or without total mesorectal excision (TME) or abdomino-perineal resection, is recommended [39]. Rectal tumors > 20 mm are automatically staged as T2 and are associated with up to 80% of distant metastases [36]. Systemic therapies are first-line in the presence of distant metastases, more for their antiproliferative effect rather than for their symptom palliation effect [39].

After curative resection, only one endoscopic follow-up at 12 months is recommended for small (<10 mm), grade 1–2 R-NETs. No further investigations are needed [39]. For grade 1–2 lesions 10–20 mm in size, an annual endoscopic follow-up is recommended with CT/MRI. For grade 1–2 lesions > 20 mm in size, a closer follow-up every 3–12 months is recommended [39].

All nine cases of R-NETs at our institution were <10 mm in size and well-differentiated. Everyone was followed for up to 1 year with repeat rectal EUS/flexible sigmoidoscopies, then continued with routine colonoscopy surveillance and had no disease recurrence. This corresponds to the non-aggressive, benign nature of the low-grade, small-sized R-NETs seen in the literature.

In conclusion, examining patients in a single healthcare system may have fewer confounders to outcomes compared to other multicenter studies. A higher incidence of GI NETs in the VA system (8.4% in our institution for the time period of 2019–2024) compared to that of the community (1%) may be due to the higher use of colonoscopies for screening and surveillance in the VA system [40]. Exposures to chemical herbicides such as Agent Orange, burn pits, and higher use of psychiatric medications may also be contributing factors to the development of NETs, although there is no medical evidence for this association yet. While there is a moderately increased risk of bladder cancer among those exposed to Agent Orange, no clinical data are available for GI NETs [41]. Additionally, our patient population consists of a higher percentage of elderly patients compared to the community, which suggests that there is more time to develop cancer. Occupational exposures such as Camp Lejeune and Gulf War syndrome could also be additional contributing factors that need further exploration, especially given the high percentage of our veterans serving in the Persian Gulf War (almost 40%) and the Vietnam War (more than 50%). While our veteran population may not be generalizable to other populations, the life expectancy of the veteran population is comparable to that of the general US population [42]. A review paper that looked at the survival of veterans with GI NETs in 1995–2009 noted that the incidence of GI NETs has increased over time in the VA system similar to non-VA settings, and both populations had similar prognoses [42]. Rectal NETs are classically known to have better survival than NETs from other primary sites, and this was observed in our population.

## 5. Conclusions

In this review, we presented a literature review of luminal GI NETs along with the GI NET cases identified at our institution from 2019 to 2024. With increased endoscopic surveillance and improvements in pathologic evaluation, the detection and treatment of GI NETs have been on the rise. The incidence of GI NETs at our institution is higher than the incidence of GI NETs in the community, and the distribution of the disease organ site regarding the order of incidence was different. However, small intestine NETs were one of the most common sites of GEP-NETs at our institution, with only one out of nineteen cases being a grade 3, poorly differentiated neuroendocrine carcinoma. This case of metastatic small bowel NET showed a poor response to both surgical and medical management with octreotide. The literature suggests that medical therapy combined with surgical intervention improves progression-free survival; however, our patient displayed a poor response, possibly due to a concurrent stage 2 colon adenocarcinoma with lymph node involvement causing a higher tumor burden. One case of E-NEC at our institution showed a complete response to chemoradiation therapy despite a significant tumor burden on presentation and high-grade pathology, perhaps due to a younger age at presentation. All nine cases of G-NETs identified at our institution also had favorable outcomes, with eight out of nine cases being type 1 and one case of metastatic G-NET showing a good initial response rate with local regional therapy with TACE and somatostatin analogs to reduce symptoms and tumor growth.

While the role of surgery or endoscopic resection is limited to localized tumors, combined treatment with chemoradiation may improve patient outcomes in high-grade, locally advanced, or metastatic diseases, as evidenced by some of the high-grade NET cases at our institution. The phase 1b KEYNOTE-028 clinical trial investigated pembrolizumab (Keytruda), a PD-1 inhibitor in patients with PD-L1-positive, locally advanced, or metastatic NETs, which showed good partial responses and good tolerance, with few adverse effects [43]. The current literature suggests that ICIs may be an effective treatment option for high-grade, poorly differentiated GEP-NENs. However, the role of immunotherapy in intermediate to well-differentiated GEP-NENs needs further exploration [44]. The increasing use of targeted therapy combined with chemotherapy is expected to provide new directions for the treatment of this rare disease.

## 6. Take-Home Points

A higher incidence of GI NETs is observed in the VA hospital system.GI NETs are a rare type of GI tumor, and improved endoscopic detection has led to an increase in GI NET diagnoses.GI neuroendocrine carcinomas have a median survival of 4 to 15 months depending on the primary site and disease stage.Surveillance guidelines following endoscopic or surgical resection depend on the stage of disease at the time of diagnosis.Long-term surveillance (up to 10 years) is imperative in patients with metastatic disease at the time of diagnosis as the majority of patients have recurrent disease.Immunotherapy, combined with targeted therapy, could represent a new treatment option for E-NECs, especially after surgical resection.

## 7. Areas of Future Research

Currently, adjuvant chemotherapy is only reserved for poorly differentiated GI NETs.The survival outcomes of patients with locally advanced E-NECs treated with definitive chemoradiation therapy need to be studied further.Further studies are needed to establish systemic treatment strategies, especially the role of immunotherapy and targeted therapies such as PRRT in locally advanced or metastatic disease.The possible factors (i.e., exposure to chemical carcinogenic substances and occupational exposures) associated with the higher incidence of NETs in veteran populations need to be further explored.

## Figures and Tables

**Figure 1 jcm-14-02148-f001:**
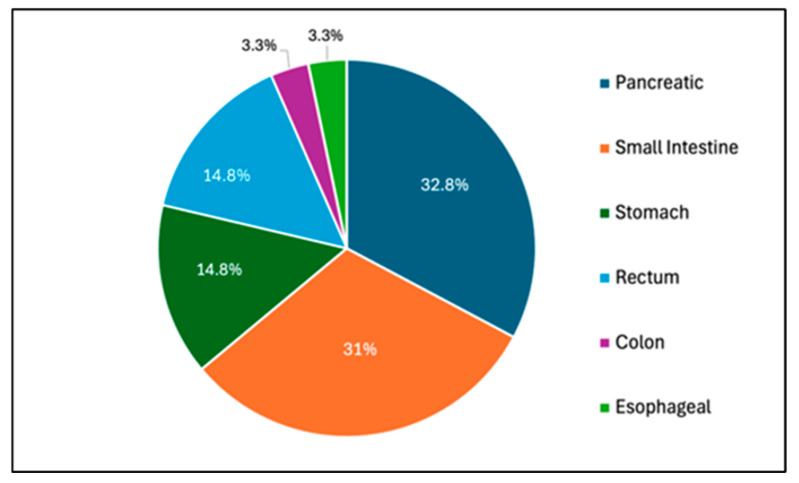
Distribution of 61 GI NET cases identified from the Hines Jr VA Hospital Cancer Registry from 2019 to 2024.

**Figure 2 jcm-14-02148-f002:**
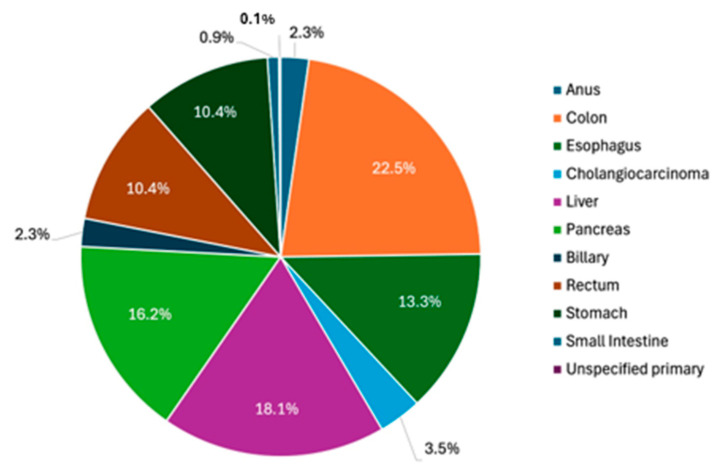
Distribution of 662 GI Tumor cases identified from the Hines Jr VA Hospital Cancer Registry from 2019 to 2024.

**Figure 3 jcm-14-02148-f003:**
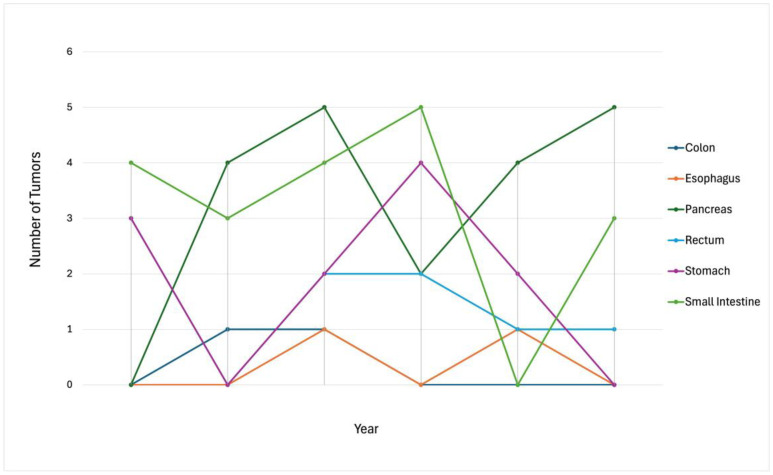
Individual GI NET incidence over a 5-year period (2019–2024).

**Figure 4 jcm-14-02148-f004:**
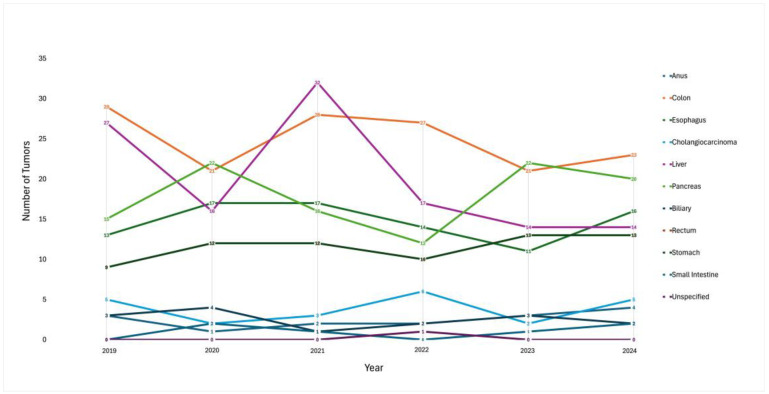
Individual GI tumor incidence by primary organ site over a 5-year period (2019–2024).

**Table 1 jcm-14-02148-t001:** Distribution of Gastric NETs.

Gastric NET Subtype	#
Type 1	8
Type 2	0
Type 3	0
Unspecified	1
	9 Total

**Table 2 jcm-14-02148-t002:** Summary of patient demographics, treatment course, and outcomes following treatment for 9 gastric NET cases.

Number	Location	Age	Gender	Race	Path/Grade	Military History	Endoscopic or Surgical Resection (Y/N)	Systemic Therapy (Chemo/RT/Immuno) (Y/N)	Medical Treatment	Surveillance
1	Stomach	77	M	White	Well-diff, no grade	Vietnam War	Y, polypectomy	N	None	Q4 months then q2 years
2	Stomach	69	M	Black	Well-diff, no grade	Vietnam War	Y, polypectomy	N	None	Annual EGD
3	Stomach	52	M	Black	Well-diff, grade 2	Persian Gulf War	Y, polypectomy	N	None	Annual EGD
4	Stomach	53	F	Black	Well-diff, grade 1	Persian Gulf War	Y, polypectomy	N	None	Annual EGD
5	Stomach	56	M	Hispanic	Well-diff, grade 2	Persian Gulf War	Y, polypectomy	N	None	Annual EGD
6	Stomach	82	M	Black	Well-to-mod diff, no grade concurrent gastric adenocarcinoma	Vietnam War	Y, Robot-assisted distal gastrectomy with roux-en-y	Y, FOLFOX	None	6-month CT AP
7	Stomach	69	M	Black	Well-diff, grade 2	Vietnam War	Y, polypectomy	N	None	Annual EGD
8	Stomach	63	M	White	Well-diff, grade 2	Persian Gulf War	N	N	Octreotide, Lanreotide, Lutathera	PET CT q6 months
9	Stomach	55	M	Hispanic	Well-diff, grade 2	Persian Gulf War	Y, Subtotal gastrectomy with roux-en-y	N	None	Annual CT/PET

**Table 3 jcm-14-02148-t003:** Summary of patient demographics, treatment course, and outcomes following treatment for 2 esophageal NET cases.

Number	Location	Age	Gender	Race	Path/ Grade	Military History	Endoscopic or Surgical Resection (Y/N)	Systemic Therapy (Chemo/RT/Immuno) (Y/N)	Medical Treatment	Surveillance
1	Esophagus	84	M	White	Poorly diff, grade 3	Korean War	N	Cisplatin/Etoposide with RT, Ipilimumab and nivolumab, Carboplatin/Etoposide	None	EGD. Patient deceased following recurrence
2	Esophagus	58	M	Hispanic	Poorly diff, grade 3	Persian Gulf War	N	Cisplatin/etoposide with RT	None	EGD with PET

While both cases of ENETs were grade 3 and poorly differentiated, there was an age difference of 26 years. Neither of them underwent endoscopic or surgical resection.

**Table 4 jcm-14-02148-t004:** Distribution of small intestinal NETs by the site of origin.

Site of Small Bowel NET	#
Duodenal Bulb	4
Jejunum	1
Ileum	1
Ileocecum	1
Distal Ileum	3
Terminal Ileum	4
Multifocal lesions	5
	19 Total

**Table 5 jcm-14-02148-t005:** Summary of patient demographics, treatment courses, and outcomes following treatment for 19 small bowel NET cases.

Number	Location	Age	Gender	Race	Path/Grade	Military History	Endoscopic or Surgical Resection (Y/N)	Systemic Therapy (Chemo/RT/Immuno) (Y/N)	Medical Treatment	Surveillance
1	Small Intestine	71	M	White	Well-diff, no grade	Vietnam War	Y, R hemicolectomy	N	Lanreotide	Hemicolectomy in 2020, did not have repeat cscope until 2025, unclear if intended for earlier.
2	Small Intestine	62	M	White	Well-diff, grade 1	Vietnam War	N	N	Octreotide, Lanreotide	Unclear; transferred care elsewhere.
3	Small Intestine	74	M	White	poorly diff, grade 3	Vietnam War	N	Y	None	Initially planned annual c-scope but stopped with poor prognosis.
4	Small Intestine	86	M	White	Well-diff, no grade	Vietnam War	Y, small bowel resection	N	Octreotide	Unclear
5	Small Intestine	80	M	White	Well-diff, grade 1	Vietnam War	Y, robotic assisted distal ileum resection	N	Octreotide	PET CT q6 months
6	Small Intestine	72	M	White	Well-diff, grade 1	Vietnam War	N	N	None	Unclear
7	Small Intestine	76	M	White	Well-diff, grade 1	Vietnam War	Y, partial duodenectomy	N	None	Q6months
8	Small Intestine	57	M	White	Well-diff, grade 2	Vietnam War	N	N	Octreotide	None. Patient died within 3 months.
9	Small Intestine	101	M	White	Well-diff, grade 1	World War II	Y, R hemicolectomy	N	None	Unclear
10	Small Intestine	73	M	Black	Well-diff, grade 2	Vietnam War	N	N	None	Q1year
11	Small Intestine	79	M	White	Well-diff, grade 2	Vietnam War	N	Y	Everolimus Lantreotide	CT AP q6month
12	Small Intestine	65	F	Black	Well-diff, grade 1	Vietnam War	N	N	None	Q3years
13	Small Intestine	44	M	White	Well-diff, grade 1	Persian Gulf War	Y, R hemicolectomy	N	None	Unclear; transferred care elsewhere
14	Small Intestine	62	M	White	Well-diff, grade 1	Vietnam War	Y, small bowel resection	N	None	Q6months
15	Small Intestine	76	M	Black	Multiple	Vietnam War	N	N	Lanreotide	PET-CT annual
16	Small Intestine	55	M	Unknown	Well-diff, grade 1	Persian Gulf War	N	N	None	Unclear
17	Small Intestine	68	M	White	Well-diff, grade 1	Vietnam War	N	N	Octreotide	CT q6month
18	Small Intestine	79	M	White	Well-diff, grade 1	Vietnam War	N	Y, capecitabine/ temozolomide	None	CT q3 months
19	Small Intestine	88	M	Black	Well-diff, no grade	Vietnam War	N	N	None	CT annual

**Table 6 jcm-14-02148-t006:** Summary of patient demographics, treatment courses, and outcomes following treatment for 2 colonic NET cases.

Number	Location	Age	Gender	Race	Stage/Grade	Military History	Endoscopic or Surgical Resection (Y/N)	Systemic Therapy (Chemo/RT/Immuno) (Y/N)	Medical Treatment	Surveillance
1	Colon	86	M	White	Well-diff, grade 2	Vietnam War	Y, Appendectomy	N	None	None
2	Colon	41	M	White	Well-diff, Low-grade	Persian Gulf War	Y, Appendectomy	N	None	None

**Table 7 jcm-14-02148-t007:** Summary of patient demographics, treatment courses, and outcomes following treatment for 9 rectal NET cases.

Number	Location	Age	Gender	Race	Stage/Grade	Military History	Endoscopic or Surgical Resection (Y/N)	Systemic Therapy (Chemo/RT/Immuno) (Y/N)	Medical Treatment	Surveillance
1	Rectum	43	F	Black	Well-diff, grade 1	Persian Gulf War	Y, endoscopic	N	None	Rectal EUS in 2 years
2	Rectum	56	M	White	Well-diff, grade 1	Persian Gulf War	Y, endoscopic	N	None	Colonoscopy in 3 years
3	Rectum	56	M	Black	Well-diff, grade 1	Persian Gulf War	Y, endoscopic	N	None	Continue surveillance colonoscopy
4	Rectum	70	M	Black	Well-diff, grade 1	Vietnam War	Y, endoscopic	N	None	Colonoscopy in 2 years
5	Rectum	46	M	White	Well-diff, grade 1	Persian Gulf War	Y, endoscopic	N	None	Continue surveillance colonoscopy
6	Rectum	60	M	White	Well-diff, grade 1	Persian Gulf War	Y, endoscopic	N	None	Continue surveillance colonoscopy
7	Rectum	53	M	Black	Well-diff, grade 1	Persian Gulf War	Y, endoscopic	N	None	Flex sig in 1 year, then sigmoidoscopy in 2 years
8	Rectum	55	M	Black	Well-diff, grade 1	Persian Gulf War	Y, endoscopic	N	None	Continue surveillance colonoscopy
9	Rectum	55	M	White	Well-diff, grade 1	Persian Gulf War	Y, endoscopic	N	None	Colonoscopy in 3 years

**Table 8 jcm-14-02148-t008:** Mortality and survival data for individual GI NETs.

Diagnosis	N (# Dead)	Mortality Rate	Average Age of Death	Treatment Type	Stage at Presentation	Stage at Death	Cause of Death	Time Between Diagnosis and Death (Months)
Gastric NET	1	11.1% (1/9)	52	Polypectomy	Well-diff, Grade 2, No metastases/ No formal staging	Well-diff, Grade 2, No metastases/No formal staging	End-stage renal disease	47 months
Esophageal NET	1	50% (1/2)	84	Chemo, RT, and immunotherapy	T4N2M0 Poorly diff, Grade 3	T4N2M0 Poorly diff, Grade 3	Recurrent disease	26 months
Small bowel NET	5	26.3% (5/19)	74	Lanreotide, Lutathera, Everolimus	Well-diff, Grade 2, No metastases/ No formal staging	Well-diff, Grade 2, No metastases/ No formal staging	Recurrent disease	4 months
				Octreotide	T4N2M1, Well-diff, Grade 2	T4N2M1, Well-diff, grade 2	Recurrent disease	5 months
				Imaging surveillance	Well-diff, Grade 1 No metastases/ No formal staging	Well-diff, Grade 1 No metastases/ No formal staging	Metastatic chondrosarcoma	20 months
				Imaging surveillance	Well-diff, Grade 1	Well-diff, Grade 1	Multiple Comorbidities	22 months
				Chemo- and immunotherapy	T3N1M1, Poorly diff, Grade 3	T3N2M1, Poorly diff, Grade 3	Recurrent disease	24 months
Colonic NET	1	50%	84	Appendectomy	Well-diff, Grade 1 No metastases/ No formal staging	Well-diff, Grade 1 No metastases/ No formal staging	Old age	39 months
Rectal NET	0	0%	N/A	Polypectomy	N/A	N/A	N/A	N/A

## Data Availability

The data supporting reported results for individual GI NET cases can be available upon request.

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
