# Peer review of "Gastroesophageal Neuroendocrine Tumors: Outcomes and Management"

_jcm, 2025, doi:10.3390/jcm14072148_

Round 1

Reviewer 1 Report

Comments and Suggestions for Authors

Manuscript entitled "Gastroesophageal Neuroendocrine Tumors: Outcomes and Management" by Christine Son et al.

This manuscript presents a retrospective review of gastroenteropancreatic neuroendocrine tumors (GEP-NETs), focusing on outcomes and management strategies.

Comments:

1.     The introduction provides a solid background on GEP-NETs but should be more concise and clinically relevant, with a more focused epidemiology discussion that reduces general details and emphasizes gaps in treatment approaches for gastroesophageal neuroendocrine tumors, while clearly defining the study’s objectives at the end—specifically, the clinical insights or unanswered questions this review aims to address.

2.     Explain whether metastatic cases were included and how patients with mixed histology (e.g., adenocarcinoma with neuroendocrine differentiation) were handled.

3.     Specify whether patients with secondary neuroendocrine tumors were included or excluded.

4.     More detailed survival analysis is needed. Compare overall survival across tumor types, treatment modalities, and metastatic vs. localized cases.

5.     Clarify how recurrence was defined and monitoredwas recurrence based on imaging, symptoms, or biomarker elevation?

6.     Expand on why GEP-NETs had a higher incidence in the Veteran population.

7.     The impact of tumor grade on treatment response should be analyzed further—do patients with low-grade vs. high-grade tumors show different responses to surgery or chemotherapy?

8.     The manuscript mentions chemoradiation therapy for esophageal neuroendocrine carcinoma (E-NEC) but does not discuss the role of novel targeted therapies (e.g., PRRT with Lutathera) or how treatment sequencing influences outcomes, such as whether systemic therapy should precede or follow surgery.

9.     The manuscript should offer clearer guidance for clinicians.

Author Response

Thank you very much for taking the time to review this manuscript. Please find the detailed responses below and the corresponding revisions in the re-submitted draft.

1. The introduction provides a solid background on GEP-NETs but should be more concise and clinically relevant, with a more focused epidemiology discussion that reduces general details and emphasizes gaps in treatment approaches for gastroesophageal neuroendocrine tumors, while clearly defining the study’s objectives at the end—specifically, the clinical insights or unanswered questions this review aims to address.

Thank you for this comment. The introduction was made more concise by moving the details of staging to results section and by clarifying study’s objectives at the end (Lines 58-61).

2. Explain whether metastatic cases were included and how patients with mixed histology (e.g., adenocarcinoma with neuroendocrine differentiation) were handled.

The patients with NET and adenocarcinoma did not have mixed histology as these were concurrent, two separate tumors. This is the unique feature we see in Veteran populations.

3. Specify whether patients with secondary neuroendocrine tumors were included or excluded.

Metastatic neuroendocrine tumors were included (Line 76).

4. More detailed survival analysis is needed. Compare overall survival across tumor types, treatment modalities, and metastatic vs. localized cases.

Table 8 was supplemented with additional survival analysis based on treatment modalities and extent of disease (Line 256).

5. Clarify how recurrence was defined and monitored—was recurrence based on imaging, symptoms, or biomarker elevation?

Recurrence was defined most frequently based on findings on routine surveillance imaging. Rarely patient presented with new symptoms, and biomarker elevation was only followed along for certain cases (Lines 262-264).

6. Expand on why GEP-NETs had a higher incidence in the Veteran population.

There is no clinical or paraclinical studies that investigate this association. However, given known chemical exposures in Veterans, we suggest such exposure may be contributed by the increased incidence of NETs in Veterans. I added a reference (#41) for the known association between Agent Orange and bladder cancer, but no studies exist for NETs. We also have higher percentage of elderly patients compared to community patients, which suggests there are more time to develop cancer. Occupational exposures such as Camp Lejeune and Gulf war syndrome could also be additional contributing factors that need further exploration (Lines 480-486).

7. The impact of tumor grade on treatment response should be analyzed further—do patients with low-grade vs. high-grade tumors show different responses to surgery or chemotherapy?

Most of the time, patients with low-grade disease did not undergo chemotherapy, so it was difficult to ascertain if patients with lower-grade disease had better responses to systemic therapy compared to patients with higher-grade tumors. However, majority of the patients who passed had higher-grade tumors (poorly differentiated, grade 2-3) – please refer to our updated mortality/survival data table 3.

8. The manuscript mentions chemoradiation therapy for esophageal neuroendocrine carcinoma (E-NEC) but does not discuss the role of novel targeted therapies (e.g., PRRT with Lutathera) or how treatment sequencing influences outcomes, such as whether systemic therapy should precede or follow surgery.

Thank you for pointing this out. I added 3 more references (#’s 25, 26, 27) to explain the role of PRRT and the impact of PRRT following surgery showing positive response (Lines 356-365).

9. The manuscript should offer clearer guidance for clinicians.

I edited “take-home points” and “areas of future research” to offer clearer guidance for clinicians.

Additional revisions:

    We have provided summary tables of patient demographics, treatment course, and outcomes following the treatment for each NET cases organized by its location. Please let us know if this is a helpful addition to the audience, as this contains similar information from the text but gives a better, quick visualization of our institutional findings.   

Reviewer 2 Report

Comments and Suggestions for Authors

This review provides a well-rounded analysis of gastroesophageal neuroendocrine tumors (NETs), covering their epidemiology, classification, treatment approaches, and patient outcomes. The inclusion of institutional data adds clinical relevance and helps contextualize the findings. Below are some key strengths and areas for improvement.

Strengths

  1. The manuscript thoroughly discusses gastroesophageal NETs, effectively integrating both literature findings and institutional data.
  2. It highlights existing knowledge gaps, particularly in managing poorly differentiated NETs, making it a useful resource for researchers and clinicians.
  3. The work is well-referenced, drawing on relevant studies to support the discussion.

Minor Comments

Some sections could be more brief or clear:

  • Lines 33–63: While the background is informative, it could be streamlined to focus more on key epidemiological statistics and clinical significance. The classification and staging details may fit better in later sections.
  • Lines 245–284: The explanation of different gastric NET subtypes is valuable but could be condensed, particularly the details on pathogenesis.
  • Lines 435–448: The conclusion should focus on key takeaways rather than reiterating all findings. A more concise summary would strengthen the impact.
  • Lines 29–31: The statement on surgery and chemoradiation improving patient outcomes should specify which tumor subtypes benefit the most.
  • Lines 456–464: It would be helpful to clarify whether institutional findings align with or differ from published literature.

More references are needed:

  • Lines 442–444: The suggestion that chemical exposures may contribute to the increased incidence of NETs in Veterans is interesting but lacks supporting citations. If relevant studies exist, citing them would strengthen this point.
  • Lines 468–470: The discussion on targeted therapy and immunotherapy would benefit from references to recent clinical trials or studies to support the claims, if any of them are available.

Overall, this review makes a valuable contribution to the field, particularly its incorporation of institutional data. Refining certain sections for clarity and conciseness, while strengthening reference support, would further enhance its readability and impact.

Author Response

Thank you very much for taking the time to review this manuscript. We are glad to hear that our institutional data and literature review provide a meaningful analysis of gastroesophageal NETs and hope this will be a useful resource for the clinicians for future areas of studies. Please find the detailed responses below and the corresponding revisions in the re-submitted draft.

  • Lines 33–63: While the background is informative, it could be streamlined to focus more on key epidemiological statistics and clinical significance. The classification and staging details may fit better in later sections.
    • The classification and staging details were moved to the results and discussion section after the incidence graphs. We clarified our study’s objectives at the end (Lines 58-61).
  • Lines 245–284: The explanation of different gastric NET subtypes is valuable but could be condensed, particularly the details on pathogenesis.
    • The explanation of different gastric NET subtypes was condensed and made more concise.
  • Lines 435–448: The conclusion should focus on key takeaways rather than reiterating all findings. A more concise summary would strengthen the impact.
    • Thank you for pointing this out. The conclusion was edited to strengthen the impact of key takeaways.
  • Lines 29–31: The statement on surgery and chemoradiation improving patient outcomes should specify which tumor subtypes benefit the most.
    • Thank you for pointing this out. I specified this is specifically for high-grade, poorly differentiated tumors.
  • Lines 456–464: It would be helpful to clarify whether institutional findings align with or differ from published literature.
    • Thank you for pointing this out. I clarified why patient with poorly differentiated grade 3 small bowel NET might have had poorer outcome than literature suggests (Lines 499-502).

More references are needed:

  • Lines 442–444: The suggestion that chemical exposures may contribute to the increased incidence of NETs in Veterans is interesting but lacks supporting citations. If relevant studies exist, citing them would strengthen this point.

·       There is no clinical or paraclinical studies that investigate this association. However, given known chemical exposures in Veterans, we suggest such exposure may be contributed by the increased incidence of NETs in Veterans. I added a reference (#41) for the known association between Agent Orange and bladder cancer, but no studies exist for NETs. We also have higher percentage of elderly patients compared to community patients, which suggests there are more time to develop cancer. Occupational exposures such as Camp Lejeune and Gulf war syndrome could also be additional contributing factors that need further exploration (Lines 480-486).

  • Lines 468–470: The discussion on targeted therapy and immunotherapy would benefit from references to recent clinical trials or studies to support the claims, if any of them are available.
    • Thank you for pointing this out. I added two references (#43 and #44) that comments on recent clinical trials and role of Immunotherapy in metastatic NETs (Lines 510-516).

Additional revisions:

    We have provided summary tables of patient demographics, treatment course, and outcomes following the treatment for each NET cases organized by its location. Please let us know if this is a helpful addition to the audience, as this contains similar information from the text but gives a better, quick visualization of our institutional findings.  

Round 2

Reviewer 1 Report

Comments and Suggestions for Authors

The authors have adequately addressed my comments, and the manuscript can be accepted for publication.